

# Impact on Cloud Properties of Reduced-Sulphur Shipping Fuel in the Eastern North Atlantic

Gerald G. Mace[1], Sally Benson[1], Peter Gombert[1], Tiffany Smallwood[1]

[1]Department of Atmospheric Science, University of Utah, Salt Lake City, UT, USA, 84112

*Correspondence to*: Gerald "Jay" Mace, Professor (jay.mace@utah.edu)

**Abstract.** The global reduction in shipping fuel sulphur that culminated in 2020 with an ~80% reduction has enabled an inadvertent experiment on the role of aerosol-cloud interaction (ACI) in the climate system. We compare observations collected at the Atmospheric Radiation Measurement program's (ARM) Eastern North Atlantic site (ARM-ENA, 39.1 N, 28.0 W) during two June to September periods: 2016-2018 (pre-2020) and 2021-2023 (post-2020). We find a significant (~15%) decrease in cloud condensation nuclei concentrations post-2020, which resulted in a decrease in cloud droplet number ($N_d$) and an increase in effective radius ($r_e$) of marine boundary layer clouds. However, cloud liquid water path (LWP) increased post-2020. The increase in LWP offset the increase in $r_e$, resulting in insignificant changes to the optical depth distribution. MODIS and CERES data in the vicinity of ENA during these periods produce similar results also with negligible change in the albedo and optical depth distributions. Regional cloud occurrence declined in line with changes in the large-scale meteorology. Our results point to a complicated interplay among the factors that modulate cloud feedback in the Eastern North Atlantic.

## 1 Introduction

Liquid clouds in the marine boundary layer (hereafter MBL clouds) are significant cooling agents in the climate system. Extended sheets of geometrically thin but optically thick stratocumulus cover broad regions of the subtropical eastern ocean basins in both hemispheres (Wood, 2012; Klein and Hartman, 1993) and influence the Earth's albedo and climate sensitivity (Klein et al., 2017). Indeed, uncertainties in simulating potential changes to MBL clouds in a warming climate contribute significantly to the uncertainty in our knowledge of the Earth's climate sensitivity (Sherwood et al., 2020). In the past decade, MBL clouds have decreased measurably and are responsible for an accelerating imbalance in the Earth's energy budget (Gossling et al., 2025). This persistent uncertainty in MBL cloud-climate interaction stems from their coupling with the large-scale atmosphere (Klein et al., 2017) and the local conditions that control MBL cloud properties when present such as the aerosol particles on which cloud droplets form (hereafter cloud condensation nuclei, CCN). This coupled system is further complicated by the occurrence of precipitation that redistributes water and heat within the MBL and removes water entirely from the atmosphere when the precipitation reaches the surface thereby influencing cloud character and coverage (Albrecht, 1989; Wood, 2005).





The amount of sunlight reaching the ocean surface relative to what is reflected to space (albedo) depends on the total droplet surface area in the vertical column that, in turn, can be related to the vertically integrated condensed liquid water mass (hereafter, liquid water path, LWP) and the number of drops per unit volume ($N_d$) within which the water is partitioned. Often a characteristic droplet size such as the effective radius ($r_e$) is used as a proxy for $N_d$ (Stephens, 1978). Twomey (1977) identified how increasing droplet surface area due to higher concentrations of CCN can influence the cloud optical depth and albedo for a given LWP.

A direct example of how anthropogenic aerosols influence marine low cloud properties and radiative effects is the ship track phenomenon that became apparent at the dawn of the satellite era (Conover, 1966). Sulphur dioxide emitted from ship stacks oxidizes into sulphuric acid aerosols to produce anomalously high CCN concentrations. Because they are often readily identifiable on satellite imagery, ship tracks have been used for decades to understand how CCN influences marine cloud albedo (e.g. Christensen and Stephens, 2012). More recently, ship tracks that are not visibly evident in imagery have been shown to produce heavily modified cloud properties (Manshausen et al. 2022). What has been more difficult to establish until recently is how shipping-related aerosol influenced clouds globally. Diamond et al. (2020) found that this source of anthropogenic CCN in marine low clouds induced a 1 W m$^{-2}$ cooling on the Earth's climate potentially offsetting a significant portion of the warming brought about by increased CO2.

A series of regulation changes from the International Maritime Organization (Osipova et al. 2024) that culminated in 2020 reduced the sulphur content of global shipping fuel from ~3.5% to 0.5%. This change has been shown to dramatically reduce the occurrence of visible ship tracks in satellite imagery (Yuan et al., 2022). Regional changes in marine low cloud microphysics associated with the IMO 2020 regulation have also been documented in a major shipping lane in the tropical Southeast Atlantic off the African continent (Diamond et al., 2023). Most recently, Yuan et al. (2024) have attempted to quantify the global impact of the IMO 2020 regulation concluding that the reduction in sulphur will produce a warming of 0.2 W m$^{-2}$. While recent studies have identified substantial reductions in marine low clouds that are directly implicated in rapid and accelerating imbalances in the Earth's energy budget (Gossling et al., 2025), the role of aerosols in these changes are a subject of debate (Gossling et al., 2025; Hodenbrog et al., 2024; Hansen et al., 2023). Yuan et al. (2024) hypothesized that this inadvertent change to marine low clouds could result in accelerated warming of the Earth. While Yuan et al (2024) suggest that some number of years will be necessary to observe the effects of the fuel change globally, regional changes such as in the heavy shipping lanes of the Eastern Atlantic may be identifiable sooner. We take up that challenge in this study.

We focus on data collected at the U.S. Department of Energy Atmospheric Radiation Measurement (ARM) Eastern North Atlantic (ENA) site located on the Portuguese Island of Graciosa in the Azores Archipelago. ARM established the ENA site in 2015 (Wood et al., 2015). We examine the warm season (June-September) when the Azores anticyclone migrates northward





and brings a higher frequency of northeasterly flows to the Azores and a high occurrence of MBL clouds over the ARM-ENA site. As detailed in Appendix A, we examine cloud properties derived from ARM data collected during periods of northeasterly flow that have been shown to have unmodified marine characteristics. We divide the data into pre-2020 (2016, 2017, 2018) and post-2020 (2021, 2022, 2023) periods (hereafter referred to as pre and post, respectively). In addition, we also examine cloud properties from those months and years derived from MODIS and CERES data in the region around the ARM site.


## 2 Results

### 2.1 Large-Scale Cloud Controlling Factors

Since the large-scale atmosphere creates the conditions for MBL cloud occurrence, we examine the extent to which several Cloud Controlling Factors (CCFs; Klein et al. (2017)) varied in the region surrounding the ARM-ENA site (Figure 1 and Table

S1 in the supplemental material). We find that aspects of the large-scale atmosphere did change measurably. While the large-scale subsidence, the mid-tropospheric humidity, and the low-level temperature advection remained nearly constant, the estimated inversion strength (EIS; Wood and Bretherton, 2006) and the near-surface wind speed distributions are measurably different. The EIS quantifies the strength of the temperature inversion that typically exists at the top of the MBL that separates the generally well-mixed and humid MBL from the drier and stratified free troposphere. The tendency for dry air to be

entrained into the MBL is a direct product of the EIS with stronger inversions resulting in less mixing and more humid MBLs with higher cloud cover. Stronger near-surface winds tend to enhance evaporation and mechanical mixing within the MBL promoting increased cloud cover (Brueck et al., 2015; Bretherton et al., 2013). The MERRA data suggest that EIS is lower in the post period while near surface winds are slightly stronger. One can imagine that these CCFs are not entirely uncoupled since more mixing would increase near surface winds.




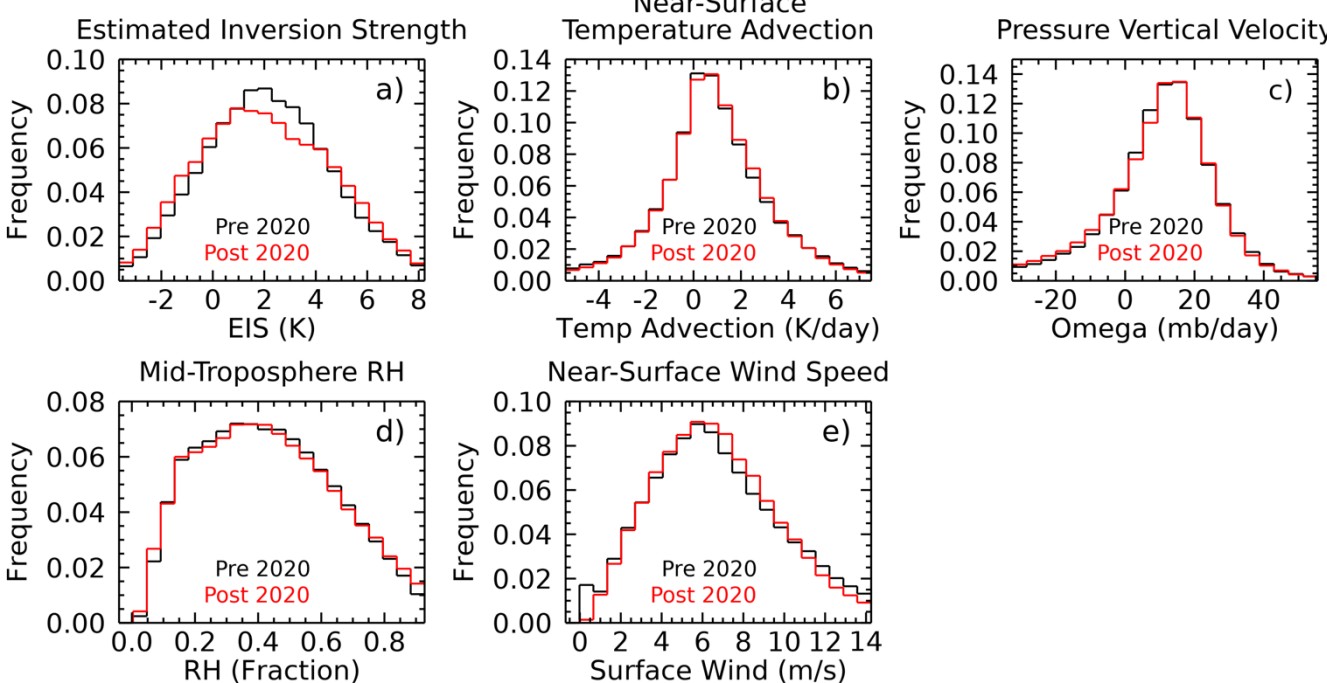

**Figure 1: Comparison of the cloud controlling factors derived from MERRA for the pre- (Black) and post-2020 (red) periods. a) Estimated Inversion Strength, b) Near-Surface Temperature Advection (K day$^{-1}$), c) pressure vertical velocity (omega, mb/day), d) mid-troposphere relative humidity (RH), and e) Near-surface wind speed (m s$^{-1}$). Statistics of these distributions are given in Table S1.**

### 2.2 ARM-ENA Observations

MBL cloud occurrence fractions at the ARM-ENA site derived from the times when the microwave radiometer provided positive LWP and the rain indicator suggested the instrument was dry were 0.787 and 0.720 in the pre and post periods, respectively. Figure 2 shows the frequency distributions of the CCN, cloud, and precipitation statistics of MBL clouds when the winds were within the directional limits for which we can be confident that the aerosol was representative of marine air during the pre and post periods. Table S2 summarizes the statistics of these distributions.

We find a significant decrease in the 0.2% supersaturation (SS) CCN concentrations from the pre to post periods. The mean 0.2% SS CCN concentrations decreased from 179 cm$^{-3}$ to 160 cm$^{-3}$, and the KS test suggests rejection of the null hypothesis that the distributions were drawn from the same sample. With a clear decrease of ~18% in the CCN concentrations, the microphysical properties of MBL clouds change in a manner consistent with Twomey (1977). We find that the $N_d$ distribution shifts significantly to lower values in the post period with the mean decreasing from 93 to 62 cm$^{-3}$ while $r_e$ increase from 12 $\mu$m to 15 $\mu$m. These changes in $r_e$ and $N_d$ would normally be associated with decreases in MBL cloud optical thickness ($\tau = \frac{3}{2\rho}\frac{LWP}{r_e}$; Stephens, 1978, where $\rho$ is the density of liquid water). However, we find that the $\tau$ distributions between the pre and





post periods are statistically indistinguishable from one another. Examining the LWP, we can see why this is so. From the pre to post periods the LWP increased from a mean of 68 g m$^{-2}$ to 73 g m$^{-2}$. The KS statistical test suggests that the null hypothesis can be just rejected at the 95% confidence interval suggesting that the LWP distributions appear not to be drawn from a similar sample population.

The formation of drizzle in marine stratocumulus clouds is strongly linked to microphysics with large cloud droplets increasing the propensity for drizzle formation (Kang et al., 2022). Defining precipitation as the occurrence of measurable radar reflectivity 100 m below the lidar identified cloud base, we find that the occurrence frequency of drizzle when clouds are present increases from 0.62 to 0.69 between the pre and post periods consistent with the increase in $r_e$. However, the distribution of drizzle rate, calculated using the regression relationship defined in Comstock et al. (2004) applied to the KAZR

data 100 m below cloud base, changes significantly with the occurrence of heavier drizzle (>1 mm day$^{-1}$), decreasing significantly in the post period.



**Figure 2.** **Cloud and precipitation derived property distributions measured during the pre (black) and post (red) periods at the ARM ENA site when wind and cloud conditions were appropriate. There are 20,990 and 15,650 5-minute samples of data in the pre and post periods, respectively. There are 187 and 134 unique days in the pre and post periods, respectively.**

## 2.3 MODIS and CERES Observations

Figure 3 and Table S3 summarize the MODIS and CERES properties from MBL cloud data collected within 250 km of the ARM-ENA site. These statistics are compiled from cloud covered pixels deemed to contain MBL clouds following the method outlined in Mace et al. (2023) and represent the properties of clouds when clouds are present. For these statistics we sample all overpasses of the region and do not filter by wind direction as was done in the ARM-ENA data. The qualitative results





comparing the pre and post periods are similar from ARM-ENA and MODIS. Recall that the MODIS bispectral algorithm retrieves the $r_e$ and $\tau$ from visible and near infrared solar reflectance measurements of 1km pixels (Platnick et al, 2003), and

the LWP and $N_d$ are inferred from the retrieved $\tau$ and $r_e$ (Grosvenor et al., 2018). The MBL cloud data suggest that $r_e$ increased while $\tau$ remained largely the same during the pre and post periods. This result implies that, like the surface-based data, the $N_d$ and LWP would have decreased and increased respectively which is what the MODIS data demonstrate. Also like the surface-based data, the $\tau$ distributions are very similar although not statistically indistinguishable at the 99% confidence level.

The Terra and Aqua satellites have the CERES instrument to directly infer the planetary albedo ($A_p$) that is defined as the fraction of sunlight reflected to space. The CERES data are measured at coarser resolution than MODIS (~20 km versus ~1 km). We find that the CERES $A_p$ distributions from the pre and post periods when clouds are present are also very similar.

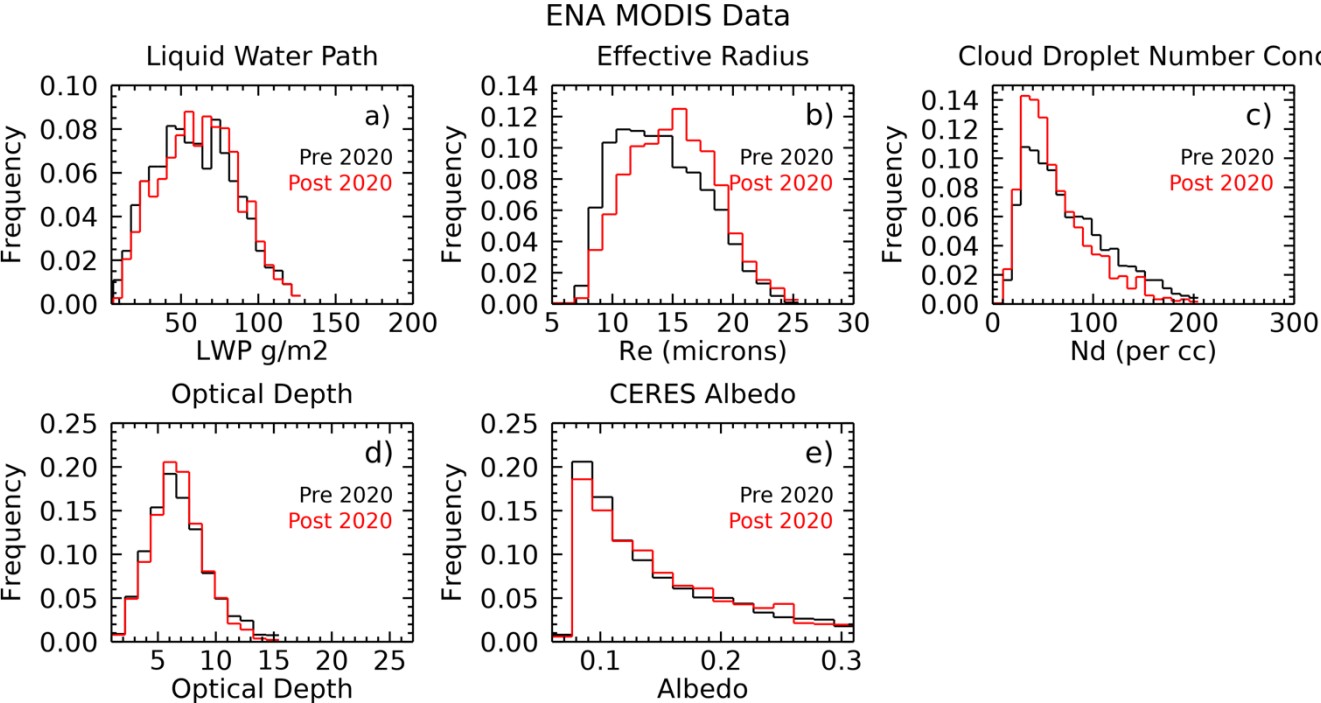

**Figure 3. MODIS cloud property distributions from data collected using the criteria listed in the Methods section within 100 km of the ARM ENA site. See the Supplemental Material for a 250 km version of these plots.**

We have been careful to note that the cloud properties we are examining are those when clouds are present. MODIS allows us to examine the occurrence frequency of low clouds. Figure 4 illustrates that the Azores Archipelago sits within a

southeasterly gradient in low cloud occurrence. The data, however, show that the pre and post periods had significantly different distributions of MBL clouds with the gradient weakening and overall cloud occurrence decreasing between the two



periods. Recall also that the ARM-ENA cloud fraction decreased from the pre to the post periods by an amount similar to that shown in Figure 4. The MODIS cloud fraction of the 1 pixel located over the ARM-ENA site decreased from 0.76 to 0.64 in the pre and post periods, respectively. These findings are consistent with the results presented in Goessling et al. (2024) who

show that the Eastern Subtropical Atlantic is within a region of increasing absorbed solar radiation anomalies and decreasing cloud cover.

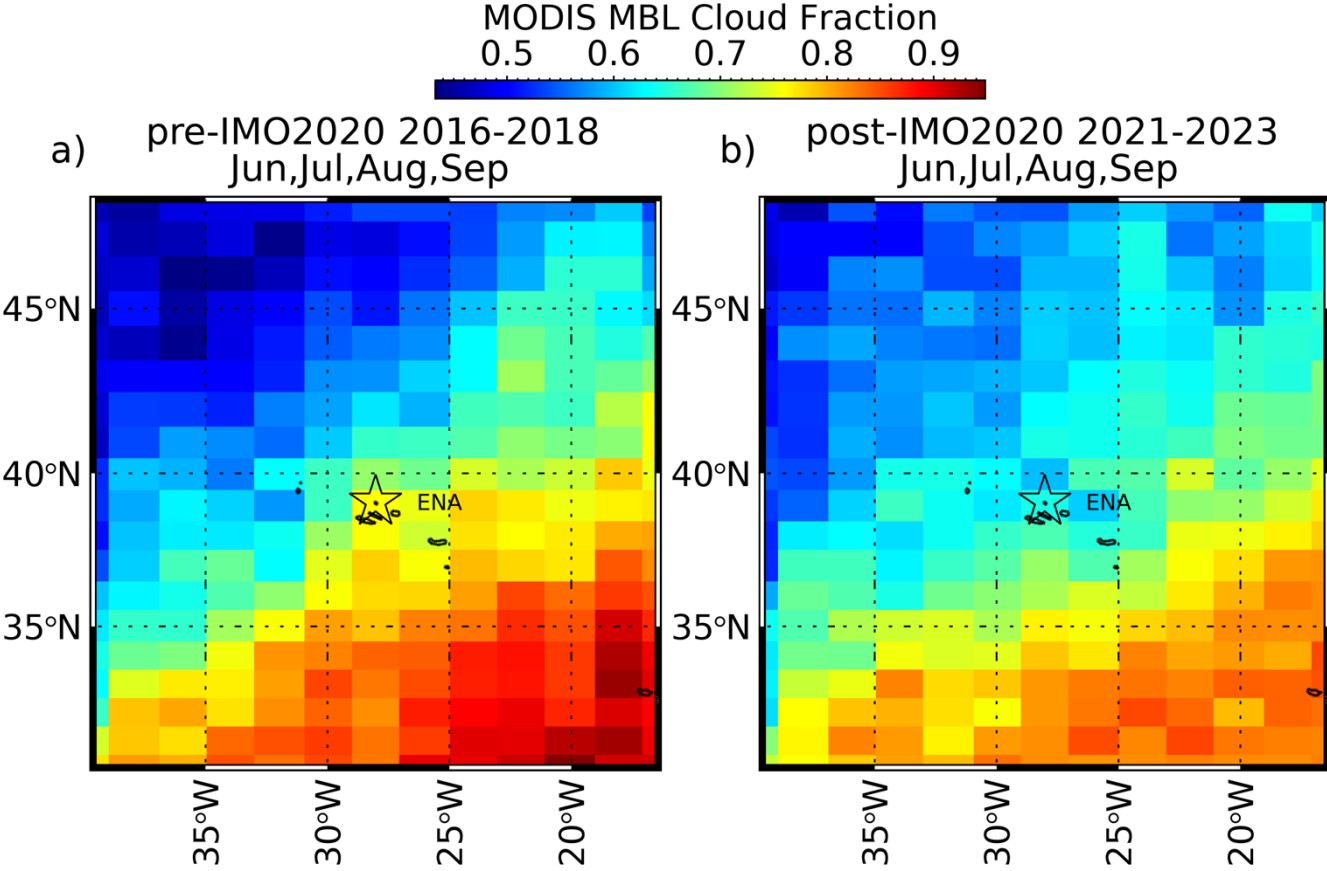

**Figure 4. Occurrence of MBL clouds during the respective years diagnosed from MODIS data. There were approximately 400 Terra and Aqua daytime overpasses of this domain during each of the pre and post periods.**


**3 Discussion and Conclusions**

The ENA region, dominated by MBL clouds during the warm season months and a major route for international shipping (Osipova et al. 2024), has undergone substantial change in the past decade. However, the aerosol changes did not occur in isolation. The large-scale atmosphere also changed during this time with the EIS becoming measurably weaker while near-





surface winds increased.  Unfortunately, the simultaneous changes in aerosol and large-scale forcing complicate any simple
conclusions that can be drawn from the surface- and satellite-based measurements of low cloud properties and occurrence.

We find both a long-term change in cloud occurrence and a microphysical response in MBL cloud properties to a decrease in
CCN that is contemporaneous with the reduction in shipping fuel sulphur content. However, both the ARM, MODIS and
CERES data suggest that the radiative effect of this change to microphysics is buffered by a slight upward adjustment to the
LWP that mostly offsets the Twomey effect when clouds are present.  LWP adjustments in MBL clouds under varying aerosol
have been reported to be both positive and negative (Chen et al., 2014; Manshausen et al., 2022; Lebsock et al., 2008; Toll et
al., 2019). The adjustments are often shown to be associated with changes to precipitation since precipitation is anticorrelated
with increases in aerosol (Toll et al., 2019).  While we find that the occurrence frequency of precipitation does increase slightly
as the $r_e$ increases, the occurrence of heavier drizzle (>0.1 mm day-1) decreases.  To determine if the change to precipitation
rate distribution was significant, we simply integrate the frequency distributions in Figure 2f recognizing that each observation
represents a 30s-time increment.  Normalizing by the cloud occurrence frequencies, we find that in the pre period the drizzle
loss to LWP by precipitation amounted to 0.159 mm per unit cloud fraction while in the post period the loss was 0.103 mm
per unit cloud fraction.  Thus, while drizzle occurs slightly more frequently in the post period with higher $r_e$, the change in
precipitation rate distribution is significant and cuts the loss of cloud water due to precipitation by a factor of ~40%.  Had the
large-scale cloud controlling factors remained unchanged, we speculate that the MBL cloud fraction would have perhaps
increased due to the reduction in loss of cloud water to drizzle perhaps resulting in a negative feedback. While this scenario
runs counter to the conclusions of Manshausen et al. (2023), such a negative feedback was hypothesized theoretically by
Stevens and Seifert (2008).


It would appear that, at least in the region near the ARM-ENA site, the aerosol impact on the radiative balance since the fuel
change in 2020 is negligible.  This conclusion is consistent with the findings of Goessling et al., (2024) in a global study who
find that the role of aerosol in the long term upward trending radiative imbalance is unclear.  Goessling et al. (2024) do,
however, link the trend in radiative imbalance to decreasing coverage of MBL clouds – especially in the Subtropical Eastern
Atlantic – also see our Fig. 4.  The change we find in the MERRA EIS is consistent with a decrease in MBL cloud cover during
the period under study.  There have been many papers based on both observations and modelling that have suggested that
increased mixing of dry air into the MBL can act to decrease cloud cover and may constitute a positive climate feedback
(Sherwood et al. 2020).  As an independent constraint on the MERRA EIS, we hypothesize that if mixing into the lower
troposphere increased, then this should be measurable as a reduction in column water vapor.  Figure 2g shows that this is
indeed the case with the column integrated water vapor measured at the ARM ENA site decreasing slightly but significantly
in the post period consistent with the idea that increased mixing of mid tropospheric dry air increased during this period due
to a weaker marine inversion.





Finally, returning to cloud properties, the question remains whether the near cancellation in the radiative response of the clouds
by offsetting changes in LWP and $r_e$ is a case of buffering in the guise of Stevens and Feingold (2009) or if the small decrease
in column water vapor and/or the weaker inversion strength acted to influence the distribution of drizzle rates. It is impossible
to answer this question with these observations. Suffice to say that changes in the Eastern North Atlantic are acting in concert
to decrease the cooling effect on the climate system imposed by MBL clouds and much work remains to understand the intricate
interactions on multiple scales that are acting to drive the climate system farther from radiative balance.


## Appendix A. Materials and Methods

We examine warm season (June-September) aerosol and cloud properties from the 2016-2019 period and compare it with data
collected in the same months from the 2021-2023 period after the IMO 2020 change was fully implemented. We refer to these
periods as "pre" and "post" periods, respectively.

**A.1 ARM Data**.

The ARM ENA site is operationally maintained with in situ and remote aerosol and cloud sensors. The ARM ENA data streams
used in this study include the following

- Cloud Condensation Nuclei: CCN were measured using the DMT CCN 100 (2016-2019) and the DMT CCN 200
  (2021-2023). The two models share identical technology with the difference being that the DMT 200 adds the
ability to measure CCN at two supersaturations simultaneously (Uin and Enekwizu, 2024). These instruments
  cycle through multiple supersaturations (SS) between 0.05 and 1 typically spending approximately 5 minutes
  collecting data at a single SS before stepping to the next typically collecting a full set of SS hourly. Because the
  CCN 100 became unreliable at SS exceeding 0.3 in 2019 and was replaced in 2021, and because the SS measured
  were slightly different during the pre and post periods, we estimate the CCN at 0.2 SS by linearly interpolating the
CCN at supersaturations below 0.3 during both periods. Mirrielees and Brooks (2018) evaluated sources of
  uncertainty in the DMT CCN instruments and found that the greatest uncertainties occurred for undercounting
  under high aerosol conditions. Such conditions are rare at ARM-ENA. Under ideal conditions Mirrielees and
  Brooks (2018) report that uncertainty in CCN concentrations are typically less than 5%.
- Cloud Radar: The Ka ARM Zenith Radar (KAZR, Widener et al., 2012) has been deployed at the ENA site since
2015. The radar collects zenith profiles of radar Doppler spectra using several operational modes designed to
  optimize detection of various hydrometeor types with a beamwidth of 0.3 degrees and time resolution of ~10
  seconds in 30 m range bins. For this study we use the General Mode that is characterized by no pulse compression
  that allows for detection of drizzle to the lowest useful range bins. For this study, we use only the zeroth moment
  of the Doppler spectrum or equivalent radar reflectivity factor. We apply a +3 dB correction factor to the radar
reflectivity as reported by Kolias et al. (2019) and assume an uncertainty in radar reflectivity factor of 3 dB.
- Microwave Radiometer (MWR): The MWR deployed at ENA is an RPG-LWP-U90 system that measures
  downwelling radiances at 23.8 and 31.4 and 90 GHz with a temporal resolution of approximately 3 seconds.





Integrated water vapor mass known as precipitable water vapor (PWV) and integrated condensed liquid water also known as liquid water path (LWP) are derived using the algorithm described by Turner et al. (2016) with an

uncertainty in LWP of approximately 20% for LWP exceeding 20 g m$^{-2}$.

- Micro Pulse Lidar (MPL): The MPL (Muradyan et al., 2020) provides copolarized and cross polarized zenith profiles of attenuated backscattered 523 nm laser light in 15 m vertical bins with a time resolution of ~10 seconds. For the methods described below we do not require calibrated attenuated backscattered measurements.

In addition, twice daily radiosonde soundings (Keeler et al., 2020) are used as well as a surface meteorological data (Kyrouac

et al., 2024) that includes wind speed and direction. To ensure that the data are minimally modified by flow over the island and the CCN are representative of marine clouds, we filter measurements to when the surface wind at the ENA Site are at least 2.5 m s$^{-1}$ and from directions between 330° and 70° and between 220° and 280° (Gallo et al., 2020).

**A.3 Cloud Properties from ARM Data:** In addition to LWP derived from the MWR, we also derive $N_d$ and $r_e$ using a method described in Mace (2024) that combines the MPL vertical profile of attenuated backscatter, the cloud boundaries from KAZR

and MPL, and the near cloud top radar reflectivity with constraints provided by the CCN measurements. This method attempts to exploit the information available from the lidar near cloud base. We find (Mace, 2024) that the vertical rate of change of the lidar signal above cloud base provides a quantitative constraint on $N_d$ when we have additional constraints on LWP from an MWR and cloud layer thickness from the combination of a cloud radar (top) and lidar (base) that takes the following form:

$$N_d = (B\eta^3\Gamma_l^2 r_{max}^5 f_{ad}^2)^{-1} \tag{A1}$$

Where $r_{max}$ is the distance from cloud base to where the vertical rate of change of the lidar attenuated backscatter changes sign. $B$ is a proportionality constant, $\eta$ is the lidar multiple scattering factor, $\Gamma_l$ is the temperature dependent adiabatic liquid water lapse rate, $f_{ad}$ is the adiabaticity of the cloud layer. The cloud top $r_e$ then follows using an equation from Grosvenor et al. (2018):

$$r_e = \left(\frac{\frac{3h}{4\pi\rho_l}\Gamma_l f_{ad}}{kN_d}\right)^{1/3} \tag{A2}$$

where $h$ is the cloud layer thickness and $k$ is the cubed ratio of the volume mean droplet radius to $r_e$. The drawback to the analytical expressions is that the results are very sensitive to our knowledge of $r_{max}$ and require a vertical resolution of better than 5m. Therefore, as described in Mace et al. (2024), we add additional information such as cloud top radar reflectivity, and lidar derived extinction ( Li et al., 2011) and cast the solution to $N_d$ and $r_e$ in terms of a Bayesian Optimal Estimation Inversion algorithm (Maahn et al., 2020) using Eqns. A1 and A2 as first guesses. This approach allows us to derive $N_d$ to within ~100%

and $r_e$ to within 30% for 30-s averaged observations. It is important to note that the derived $N_d$ is not entirely independent of the CCN since we use the CCN concentration as an upper constraint to the inversion algorithm. We further restrict our analysis of the ARM ENA microphysical retrievals to when cloud base and top are less than 4 km above the surface and the LWP is greater than 20 g m$^{-2}$ and the "rain flag" indicates that the MWR instrument was dry.

**A.4 Satellite Data**



We also examine cloud properties derived from MODIS instruments on the Terra and Aqua satellites when they pass within 500 km of the ARM ENA site (Platnick et al., 2015a, b). The MODIS algorithm uses reflected sunlight in visible and near infrared spectral bands to derive the optical depth ($\tau$) and $r_e$ (Nakajima and King, 1990) from which LWP and $N_d$ are derived (Grosvenor et al. 2018). Our approach to compiling MODIS cloud property statistics is described in Mace et al., (2023) where we restrict analysis to ice-free MBL cloud scenes with LWP<300 g m$^{-2}$ to avoid drizzle that complicates the retrieval (Xu et

al., 2021). Furthermore, we restrict our analysis to view zenith angles less than 30. Our analysis is restricted to the June-September periods of the years considered for the ARM ENA data (2016-2019 and 2021-2023). In Mace (2024) and Mace et al. (2024) we present comparisons of MODIS and surface-based retrievals of $N_d$, $r_e$, LWP, and $\tau$. In addition, we use the observed collocated MBL cloud albedo from the Clouds and the Earth's Radiant Energy System (CERES) Energy Balanced and Filled (EBAF) v 4.0 (Loeb et al, 2018) data from instruments on the Terra and Aqua satellites that are coincident with the

MODIS MBL cloud scenes. For this analysis we have ~400 MODIS passes during each 3-year period that provide data within 500 km of the ARM ENA site.

**A.5 Large-Scale Meteorology**

The large-scale meteorology was obtained from the Modern-Era Retrospective analysis for Research and Applications 2 (MERRA2) product (GMAO, 2015). We focus on the cloud controlling factors (CCFs) highlighted in Klein et al. (2017) and

examine the Estimated Inversion Strength (EIS, Wood and Bretherton, 2006), free tropospheric subsidence, cold air advection, free tropospheric humidity, sea surface temperature, and surface wind speed. Like the surface-based and satellite cloud data, we compare the CCFs within 500 km of the ARM ENA site for the warm season months (May-September) and in the 2016-2019 and 2021-2023 periods. We make no attempt to subsample the MERRA2 data in the region for the presence of MBL clouds.

**A.6 Statistical Significance Testing**

We compare frequency distributions of quantities observed and calculated from the pre and post periods. Our goal is to determine whether the distributions changed significantly. To quantify this evaluation, we use Kolmogorov–Smirnov (KS) statistical tests as described in Peacock, (1983) and Lopes et al., (2007). The KS statistic uses differences in the cumulative probability distributions of two samples to quantify the likelihood that the two samples are drawn from the same population.

It can be shown that if two samples are drawn from the same population, then the maximum difference $D_{max}$ in their cumulative distribution functions is expected to be $\frac{1}{\sqrt{N_e}}$ where $N_e$ is the effective number of independent measurements. The sampling distribution of the test statistic $Z_{Sim} = D_{max}\sqrt{N_e}$ is well known for large $N_e$ and allows for a determination of the probability $p$ that $Z_{sim}$ is greater than a value of $Z$ derived from two measured distributions. As $p$ becomes larger it is increaslingly likely that the two measured distributions are drawn from the same population (the null hypothesis). As commonly implemented

(Press et al. 1992), we reject the null hypothesis when p < 0.01 (i.e., 99% confidence) and infer that the two distributions cannot be claimed with certainty to have been drawn from the same population. Because the number of independent samples, $N_e$, is an important but potentially ill-defined parameter, we assume that measurements collected during a particular day or



during a particular overpass of the satellite occurred within the same large-scale regime and were likely not necessarily independent. Therefore, $N_e$ is taken to be the average number of MODIS overpasses in the pre and post periods for the MODIS data (400 overpasses in each period), the average number of sampled days for the ARM ENA periods when clouds were present from the appropriate wind direction (150 days) and days of MERRA reanalysis data in each period (600). For the LWP and PWV data presented in Fig. 2, we did not filter by wind direction and therefore use the number of individual days in the pre and post periods.

**Author Contributions**.

GM led the overall conception, data analysis, interpretation of results, and writing of the text of the article. SB was responsible for implementing data analysis code and generation of figures. TS and PG contributed to data analysis and interpretation.

**Data and Code Availability**:

All data sets used in this study were acquired from public archives and are fully cited in the text with DOIs given in the reference section. Code and interim file availability is being arranged at the time of submission and are pending based on reviewer comments and eventual modification based on reviewer comments. When appropriate, we will use the data repository at the University of Utah (https://hive.utah.edu/?locale=en) as we have done for other recent peer review articles.

**Competing Interests**:

The authors declare no competing interests.

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
