# Peer review of "Impact on Cloud Properties of Reduced-Sulphur Shipping Fuel in the Eastern North Atlantic"

_EGUsphere, 2025_

## Author Response (AR1)

We would like to thank the reviewers for their insightful comments. We have endeavored to address each of them to the extent possible. Below, we list the reviewer comments as indents followed by our responses in italics.

- **RC1**: 'Comment on egusphere-2025-2075', Anonymous Referee #1, 26 Jun 2025  reply

  Mace et al. are interested in the impact of the shipping sulfur regulations imposed by the MLO in 2020 on cloud properties and the Earth radiation budget. They present a concise and useful analysis of systematic ground-based remote sensing at a maritime observatory station (yet another manifestation of the outstanding usefulness of the American Atmospheric Radiation Measurement Programme that around the world we would very much like to see continued). They complement this with an analysis of satellite retrievals and meteorological reanalyses.

  The study is very well written, grounded in a concise and knowledgeable report about the state of the art. The results are diligently discussed. I only have a few minor suggestions for consideration as minor revisions.

  - l33 Not quite the exact definition of albedo, which rather is the ratio of reflected to incoming solar radiation

  *Response: Changed to formal definition in text.*

  - l45 This statement is a bit misleading. Diamond et al. estimate the -1 Wm-2 when considering all anthropogenic aerosol sources, not just the ones from shipping.

  *Response: Appreciate catching this. I added a more concise statement in the text that correctly represents the Diamond et al. finding.*

  - l88 why not also the unit for EIS in the caption

  *Response: Added as suggested.*

  - l92 Surface temperature is of course not a cloud-controlling factor, but it might still be interesting to also show it here. The reason is that warming in between pre and post might explain aspects of the cloud changes, such as the LWP increase

  *Response: This is an interesting idea, and I have considered it. The SST's are higher in the post period by ~0.3 K suggesting that there may be a potential link to higher LWP. However, I think the reason that surface temperature is not a cloud controlling factor is that the responses of clouds to temperature change are non-monotonic. Higher SST would increase the potential for there to be more absolute moisture in the MBL and the adiabatic liquid water content would increase due to a weaker moist adiabatic lapse rate.*

*On the other hand, temperature tends to weaken EIS (what we find). SST and EIS are certainly coupled and would result in a drier MBL. The literature points to both responses but seems to come down on the side of the EIS-related drying being predominant. The study by Gordon and Klein (2014) addresses this issue directly using models and the results of Eitzen et al. (2011). They conclude that the EIS-related drying due to warming is predominant in warm stratocumulus regions. This is also consistent with the analysis of Sherwood et al. (2014). As a matter of fact, the global changes to low clouds over the past two decades are consistent with this positive feedback modulated by other factors such as ocean mixing. I now note with a new paragraph in section 2.1 that modest increases in SST are observed between pre and post, but the effect of this temperature increase is likely more coupled to the decrease in EIS citing the literature above.*

*References:*

*Eitzen, Zachary A., Kuan-Man Xu, and Takmeng Wong. "An Estimate of Low-Cloud Feedbacks from Variations of Cloud Radiative and Physical Properties with Sea Surface Temperature on Interannual Time Scales." Journal of Climate24, no. 4 (February 15, 2011): 1106–21. https://doi.org/10.1175/2010jcli3670.1.*

*Gordon, Neil D., and Stephen A. Klein. "Low-cloud Optical Depth Feedback in Climate Models." Journal of Geophysical Research: Atmospheres 119, no. 10 (May 27, 2014): 6052–65. https://doi.org/10.1002/2013jd021052.*

*Sherwood, S. C., M. J. Webb, J. D. Annan, K. C. Armour, P. M. Forster, J. C. Hargreaves, G. Hegerl, et al. "An Assessment of Earth's Climate Sensitivity Using Multiple Lines of Evidence." Reviews of Geophysics 58, no. 4 (September 25, 2020). https://doi.org/10.1029/2019rg000678.*

o l100 is the change by 19 cm-3 relative to 160 cm-3 not 12%, rather than 18%?

*Response: Changed to correct math. Thanks for catching.*

o l102 any idea why the change by 50% is much larger than the one for CCN?

*Response: I noticed this also. I don't have an explanation that is more than hand waving. The change in MODIS Nd is closer to the change in CCN. I would point perhaps to uncertainties in both estimates of Nd and to the fact that Nd is controlled by more than CCN number being sensitive to precipitation (the clouds are drizzling more) and hygroscopicity of the aerosol in the actual updrafts (the reduction of sulfate would have shifted the overall hygroscopicity probably more towards that of sea salt). Also, the MODIS and surface data see different parts of the cloud (top versus base, respectively). The surface would be more sensitive to the increase in precip occurrence while the top of the layer less so. I note the Nd change discrepancy in the first paragraph of section 2.3*

*but provide no real discussion of it since it would be speculation to delve into the topic more deeply.*

- o 1104 maybe it is worth noting this scaling is for vertically uniform droplet size distributions

*Response: Noted this in the Appendix where we discuss method. The assumption of vertically invariant Nd is true for both the surface and satellite retrievals. While there is certainly variability in the real world, the assumption is reasonably consistent with aircraft data as noted in several observational studies.*

- o 1106 it might be interesting to examine (e.g. in Fig. 1) the temperature changes. Is perhaps the LWP increase (partly) consistent with an increase in adiabatic liquid water content in response to warming? if so, one might be able attempt to deconvolve the warming from the aerosol aspects

*Response: See my response regarding your l92 comment.*

- o 1113 very interesting
- o 1170 just to clarify – this would be the expected signal. The accumulated precipitation over all intensity classes should change only in response to surface and atmospheric energy budget changes.

*Response: Correct that increasing precipitation occurrence is expected with an increase in effective radius. I note this now in that sentence.*

- o 1173 I am not sure I understand this metric. first of all, where is the time unit, does one not need a rate? Second, why per horizontal cloud fraction and not rather something more related to LWP? Maybe a formula would help. Or maybe this calculation does actually not help the understanding.

*Response: I appreciate that the motivation and method for the precipitation analysis was not fully described. I have added the following text to the Methods Appendix. Hopefully this conveys the information more clearly.*

*Our objective is to compare how the clouds in each period lost water to precipitation, $q_{loss}^{period}$, where period refers to pre and post with units of mm per unit cloud fraction. Each observation of precipitation (P) is given as a rate (mm/sec, let's say). I have some number of occurrences (n) of P in some number (N) of precipitation rate bins ($P_{bins}$). If each observation of P represents a 30 second interval (dt), then each observation of P\*dt would have units of mm, and summing all the observations in each rate bin ($P_{bins}$) would have units of mm of water lost from the clouds at that rate. In other words, simply summing $\sum_n P dt$ approximates the total water in mm lost to precipitation in that P bin. Summing across the N $P_{bins}$ bins gives the total number of mm of precipitation that the frequency distribution represents, $Q_{loss}^{period}$. In other words*

$$Q_{loss}^{period} = \sum_{N} \sum_{n} P dt$$

*However, to compare the efficiency at which water is lost to precipitation between the two periods, pre and post, to evaluate which loses more water to precipitation in a relative sense, we find it instructive to normalize by the cloud occurrence, f of a period, $f^{period}$. Let's say that there were twice as many clouds in the post period as the pre period, but the precipitation rate frequency distributions were the same, then $Q_{loss}^{post} = 2Q_{loss}^{pre}$. In our case, we have less clouds in the post period but more precipitation overall and comparing $Q_{loss}^{post}$ with $Q_{loss}^{pre}$ would be ambiguous without some normalization. So, normalizing $Q_{loss}^{period}$ by the cloud fraction, f, of that period, allows us to compare the efficiency with which clouds in each period lost water to precipitation relative to the other, or*

$$q_{loss}^{period} = \frac{Q_{loss}^{period}}{f^{period}}$$

**RC2**: 'Comment on egusphere-2025-2075', Mark Miller, 03 Jul 2025

- Line 54: "produce a warming of 0.2 W m⁻²: suggest "produce an increase in global surface radiation of 0.2 W m⁻².

*Response: Changed as suggested.*

- Line 81: Stronger surface winds may also lead to increased mesoscale organization in convective clusters.

*Response: This is a very interesting point. I think it is a bit beyond the scope of the present manuscript, but we note it for future work. I've added the following text and references to section 2.1:*

*The increase in surface winds has the potential to modify the mesoscale organization of the marine stratocumulus fields. (Wood & Hartmann, 2006; Stevens et al., 2005; Wang & Feingold, 2009; Yamaguchi & Feingold, 2015). Increases in wind speed can enhance turbulent fluxes and drizzle, occasionally promoting transitions toward more open or organized cellular convection. However, the ENA warm-season clouds analysed here generally occur in moderate-wind conditions below the threshold where such mesoscale transitions are pronounced. We therefore expect mesoscale organization to have limited influence on the observed cloud property changes. However, exploring this further would be an interesting topic of follow up studies.*

*References:*

*Wood, Robert, and Dennis L. Hartmann. "Spatial Variability of Liquid Water Path in Marine Low Cloud: The Importance of Mesoscale Cellular Convection." Journal of Climate 19, no. 9 (May 1, 2006): 1748–64. https://doi.org/10.1175/jcli3702.1.*

*Stevens, Bjorn, Gabor Vali, Kimberly Comstock, Robert Wood, Margreet C. van Zanten, Philip H. Austin, Christopher S. Bretherton, and Donald H. Lenschow. "Pockets of Open Cells and Drizzle in Marine Stratocumulus." Bulletin of the American Meteorological Society 86, no. 1 (January 2005): 51–58. https://doi.org/10.1175/bams-86-1-51.*

*Wang, Hailong, and Graham Feingold. "Modeling Mesoscale Cellular Structures and Drizzle in Marine Stratocumulus. Part I: Impact of Drizzle on the Formation and Evolution of Open Cells." Journal of the Atmospheric Sciences 66, no. 11 (November 1, 2009): 3237–56. https://doi.org/10.1175/2009jas3022.1.*

*Yamaguchi, T., and G. Feingold. "On the Relationship between Open Cellular Convective Cloud Patterns and the Spatial Distribution of Precipitation." Atmospheric Chemistry and Physics 15, no. 3 (February 5, 2015): 1237–51. https://doi.org/10.5194/acp-15-1237-2015.*

- o Figure 2: Smaller effective radii, increased CCN, pre 2020 but increased precipitation relative to post 2020 seems counterintuitive. Does this suggest that LWP, which is greater in the pre-2020 period, is a stronger modulator of precipitation?

*Response: We find that the occurrence of precipitation increases post 2020 with larger effective radii but in looking more closely at how the distribution of precipitation rates change, we conclude that the change in the distribution of precipitation results in less loss of cloud water overall in the post 2020 period. I agree that this is counter to intuition. I've added some explanatory text to the methods describing our approach to analyzing the drizzle rate changes.*

- o The 2021-2023 corresponds to an extended, strong La Nina. As you note in Line 160, changes in large scale circulation complicate any simple conclusions, but you might consider mentioning a possible link between the ENSO state and cloud coverage over the ENA.

*Response: This is a very interesting point. The post-2020 analysis period (2021–2023) coincided with a prolonged La Niña event. Although the ENA region lies well outside the tropical Pacific ENSO centers, teleconnections from La Niña can influence North Atlantic large-scale circulation and surface temperature patterns. These changes could indirectly affect marine low clouds through modulations in subsidence due to anomalously high geopotential heights (Knight and Scaife, 2024). However, we find no measurable difference in subsidence between the pre and post periods (Fig. 1). While other such influences may still contribute to the observed large-scale differences between the pre- and post-IMO periods, the amplitude of these effects over the ENA is likely modest relative to the local variations in EIS and aerosol conditions identified here.*

*Furthermore, I would expect such anomalies to enhance the EIS which is opposite to what we find.*

*References:*

*Knight, Jeff R., and Adam A. Scaife. "Influences on North-atlantic Summer Climate from the El Niño-southern Oscillation." Quarterly Journal of the Royal Meteorological Society 150, no. 764 (August 7, 2024): 4498–4510. https://doi.org/10.1002/qj.4826.*

- o Line 170: "While we find that the occurrence frequency of precipitation does increase slightly as the $r_e$ increases, the occurrence of heavier drizzle (>0.1 mm day-1) decreases". Heavy drizzle at ENA typically occurs when cumulus-coupled stratocumulus is present, hence this decrease in heavier precipitation may portend a reduction in cumulus-coupling events.

*Response: While this point seems interesting, we did not attempt to analyze the cloud and boundary layer structure at this level of detail. This would be an excellent topic for future work and we note this in the conclusions.*

- o Line 176: "we speculate that the MBL cloud fraction would have perhaps increased due to the reduction in loss of cloud water to drizzle perhaps resulting in a negative feedback." If drizzle evaporation in the sub cloud layer is reduced, sub cloud layer stability is decreased, which may also help to promote an increase in cloud fraction.

*Response: During the post period, light drizzle increased in frequency consistent with an increase in effective radius. The heavier drizzle decreased but the heavier drizzle was more likely to reach the surface instead of evaporating completely in the MBL. So, perhaps the decrease in cloud fraction that we observed in the post period was, in part, influenced by this effect. I think this fits into my final statement in the conclusions: "much work remains to understand the intricate interactions on multiple scales that are acting to drive the climate system farther from radiative balance."*

- o Line 195: "or if the small decrease in column water vapor and/or the weaker inversion strength acted to influence the distribution of drizzle rates". As I noted in an earlier comment, at a process level, a change in the mode of the ENA cloud structure could also influence the distribution of drizzle rates over the ENA.

*Response: Quantifying changes to the mesoscale structure remains a topic of future work and is beyond the scope of the present study. Hopefully, interested researchers can pick up this topic and drill more deeply into the role of the various factors. I note this in the final paragraph of the conclusions section.*